# Identification of a Novel Oxidative Stress- and Anoikis-Related Prognostic Signature and Its Immune Landscape Analysis in Non-Small Cell Lung Cancer

**DOI:** 10.3390/ijms242216188

**Published:** 2023-11-10

**Authors:** Hanqing Zhao, Ying Huang, Guoshun Tong, Wei Wu, Yangwu Ren

**Affiliations:** Department of Epidemiology, School of Public Health, China Medical University, Shenyang 110122, China; zhq2129576286@163.com (H.Z.); huangying980131@163.com (Y.H.); 2021120138@cmu.edu.cn (G.T.)

**Keywords:** oxidative stress, anoikis, NSCLC, TME, immune landscape, stemness

## Abstract

The objective of this study was to identify a kind of prognostic signature based on oxidative stress- and anoikis-related genes (OARGs) for predicting the prognosis and immune landscape of NSCLC. Initially, We identified 47 differentially expressed OARGs that primarily regulate oxidative stress and epithelial cell infiltration through the PI3K-Akt pathway. Subsequently, 10 OARGs related to prognosis determined two potential clusters. A cluster was associated with a shorter survival level, lower immune infiltration, higher stemness index and tumor mutation burden. Next, The best risk score model constructed by prognostic OARGs was the Random Survival Forest model, and it included SLC2A1, LDHA and PLAU. The high-risk group was associated with cluster A and poor prognosis, with a higher tumor mutation burden, stemness index and proportion of M0-type macrophages, and a lower immune checkpoint expression level, immune function score and IPS score. The calibration curve and decision-making curve showed that the risk score combined with clinical pathological characteristics could be used to construct a nomogram for guiding the clinical treatment strategies. Finally, We found that all three hub genes were highly expressed in tumor tissues, and LDHA expression was mainly regulated by has-miR-338-3p, has-miR-330-5p and has-miR-34c-5p. Altogether, We constructed an OARG-related prognostic signature to reveal potential relationships between the signature and clinical characteristics, TME, stemness, tumor mutational burden, drug sensitivity and immune landscape in NSCLC patients.

## 1. Introduction

According to the latest global cancer statistics, lung cancer is still the leading cause of cancer death. In 2020, 1.8 million people died of lung cancer, accounting for 18% of all cancer deaths [1]. Among lung cancer, non-small cell lung cancer (NSCLC) accounts for about approximately 85% [2] and is the primary focus of lung cancer research. NSCLC has a high rate of recurrence and metastasis; even in cases of surgery and chemotherapy, about 65% of the patients still experience disease persistence or distant metastasis, and the 5-year survival rate is less than 10% [3]. Therefore, there is an urgent need for more novel biomarkers to predict the prognosis of NSCLC patients so that timely clinical interventions can be taken to delay the progression of the disease.

Anoikis is a type of programmed cell death that occurs when cell–cell or cell–matrix junctions are disrupted. It plays a crucial role in maintaining the dynamic balance between normal cell proliferation, differentiation and apoptosis [4,5,6]. Therefore, anoikis can eliminate circulating tumor cells (CTC), inhibit tumor cell proliferation and ectopic transplantation, and ultimately inhibit tumor metastasis [7,8]. Conversely, anoikis resistance allows isolated cells to bypass death signaling pathways, which becomes a prominent feature of cancer cell distant metastasis [9,10].

Oxidative stress (OS) refers to cytopathological changes caused by the exposure of cells to high concentrations of oxygen or oxygen derivatives [11]. It is a state of imbalance between the normal oxidant-scavenging enzyme system and the production of intracellular reactive oxygen species (ROS) [12]. When the oxidative stress state is more serious, this will result in a variety of key normal cells in the change in the macromolecular structure and function, which can lead to DNA damage, molecular mutation and chromosome instability and signaling pathway change, eventually leading to the formation of a tumor [13,14]. Although oxidative stress can promote the occurrence of tumors, long-term large amounts of ROS also have toxic effects on malignant tumors [15], and the effect of many anti-cancer methods depends on their ability to promote oxidative stress [16]. In addition, many studies have shown that oxidative stress is closely related to programmed cell death (such as autophagy, ferroptosis, cuproptosis), and oxidative stress can induce or activate programmed cell death [17,18,19,20,21].

Anoikis, as a programmed cell death mode, is also closely related to oxidative stress. Anoikis induces cell death mainly by interfering with the endogenous pathway of mitochondria or activating the exogenous pathway of death receptors on the cell surface [22,23]. Mitochondria are the main targets of intracellular ROS [24], and ROS-induced oxidative stress or damage to the antioxidant system can result in mitochondrial dysfunction and initiation of the cell death cascade [25], triggering anoikis. Anoikis resistance in matrix-isolated breast cancer cells has also been found to be induced by p-cadherin through reducing oxidative stress [26]. While the relationship between oxidative stress and anoikis has been extensively studied in various cancers [27,28,29], there is limited research on this topic specific to lung cancer. Therefore, exploring the role of oxidative stress- and anoikis-related genes (OARGs) in non-small cell lung cancer may help identify novel therapeutic strategies for NSCLC.

With the advent of the artificial intelligence era, machine learning techniques are gaining increasing importance in bioinformatics. Machine learning methods possess the capability to automatically identify patterns in data, integrate multiple heterogeneous features, and construct highly complex models required for certain genomic elements [30]. The Least Absolute Selection and Shrinkage operator (LASSO) is a regression algorithm that utilizes the L1 penalty to shrink the regression coefficients of x variables that have little or no influence on the dependent variable to zero [31], resulting in better sparsity of the regression coefficient vector. Random Survival Forest (RSF) is considered a more accurate method for handling right-censored survival data. It builds multiple decision trees using the bootstrap method to predict outcomes [32], allowing for the modeling of nonlinear effects and complex interactions between factors [33]. Gradient Boosting machines (GBM) focus on improving the prediction of a machine learning algorithm by combining variables that may not be individually important but are highly informative when considered together [34].

In this study, we identified relevant OARGs in non-small cell lung cancer, evaluated their expression levels, genetic alterations, enriched pathways and functions, immune landscape, stemness features and prognostic value in NSCLC. Additionally, we developed three OARG-based risk score models based on machine learning. The best Random Survival Forest model was selected for subsequent analysis.

## 2. Results

### 2.1. Identification of OARGs Associated with Prognosis

In total, 185 OARGs were obtained by intersecting oxidative stress-related genes and anoikis-related genes using a Venn plot (Figure 1A). Furthermore, 47 DEGs were identified from the TCGA data set (TCGA-LUAD and TCGA-LUSC) by comparing the expression between normal tissues and NSCLC tissues. (Figure 1B). GO and KEGG analyses were performed to explore the differentially expressed OARG-related biological processes (BP), cellular components (CC), molecular functions (MF) and pathways (Figure 1C,D). The results showed that these OARGs are mainly associated with the following functions and pathways: BP: cellular response to chemical stress, response to oxidative stress, and epithelial cell proliferation; KEGG: PI3K-Akt signaling pathway and MicroRNAs in cancer. This indicates that OARGs are closely related to oxidative stress and epithelial cell infiltration, and regulate these processes through the PI3K-Akt and MicroRNAs pathways. Finally, univariate COX regression analysis identified 10 prognostic OARGs significantly associated with poor prognosis (Figure 1E, Appendix A).

### 2.2. Analysis of Related Characteristics of Prognostic OARGs

A network diagram was used to demonstrate the relationship between the expression levels of these 10 prognostic OARGs (Figure 2A). All genes showed a positive correlation with each other, except THBS1 and BIRC5. Copy number variation (CNV) analysis of OARGs in NSCLC revealed that NDRG1, BIRC5, SLC2A1, PLAU, CDKN3, BRCA1, and SPP1 exhibited increased CNV, while PCNA, LDHA, and THBS1 showed decreased CNV (Figure 2B). The location of CNVs in prognostic OARGs on human chromosomes was also illustrated (Figure 2C). SLC2A1 is located on chromosome 1, SPP1 on chromosome 4, NDRG1 on chromosome 8, PLAU on chromosome 10, LDHA on chromosome 11, CDKN3 on chromosome 14, and THBS1 on chromosome 15. BIRC5 and BRCA1 are both located on chromosome 17 and PCNA on chromosome 20. Furthermore, the incidence of somatic mutations in prognostic OARGs in NSCLC patients was calculated, with BRCA1 showing the highest mutation frequency dominated by Missense_Mutation. (Figure 2D). Moreover, NDRG1 and SLC2A1 are significantly mutated (*p* < 0.05) (Figure 2E,F).

### 2.3. Consensus Clustering of Potential Molecular Clusters of OARGs

According to the expression level of OARGs, 1193 NSCLC patients were divided into two clusters by the consensus clustering method (Figure 3A). The eight curves of the consistency Cumulative Distribution Function (CDF) plot represent the cumulative distribution function of the consistency matrix for different values of k (number of clusters). The figure showed that when k was 2, the CDF took the maximum value, which proves the correctness of the result of the consistency matrix (Figure 3B). PCA (Principal Component Analysis) was performed to separate the clusters to verify the effect of clustering. The A in the PCA plot (Figure 3C) corresponded to the 1 in the consistency matrix (Figure 3A), and B corresponded to the 2 in the consistency matrix (Figure 3A). Because 1 accounted for most of the matrix in the consistency matrix, this meant that it had a larger sample size, and cluster A accounted for most of the samples of non-small cell lung cancer. (A accounted for 714 samples and B accounted for 479 samples). as shown in Figure 3C. Kaplan–Meier (KM) analysis showed that cluster B had a longer survival than cluster A (*p* < 0.001) (Figure 3D). The heatmap displayed the expression levels and clinical characteristics of OARG-related clusters in NSCLC patients, revealing that except for THBS1, all prognostic-related OARGs were highly expressed in cluster A, which was associated with poor prognosis (Figure 3E). Moreover, males and squamous cell carcinoma patients were more clustered in cluster A, indicating that the high expression of these genes in male and squamous cell carcinoma is more likely to lead to a poor prognosis.

### 2.4. Analysis of Related Characteristics of OARG Subgroups

The boxplot showed the expression profile of OARGs (including BIRC5, CDKN3, PLAU, SLC2A1, LDHA, SPP1, NDRG1, PCNA and BRCA1) in the two clusters. All the given genes are significantly upregulated in cluster A compared with cluster B (Figure 4A), corroborating the findings from the heatmap (Figure 3E). It also justified the conclusion that cluster A has a shorter survival period than cluster B (Figure 3D). This is because the expression of these genes is a risk factor for the prognosis of NSCLC patients (Figure 1E). Moreover, these genes were all highly expressed in cluster A, indicating that cluster A was associated with poor prognosis and shorter survival. The results of ssGSEA showed that most of the immune cells, such as activated B cells, activated CD8 T cells, natural killer cells, and activated dendritic cells, were more infiltrated in the B cluster (Figure 4B, *p* < 0.05). Furthermore, GSVA and GSEA methods were used to compare the related enrichment pathways and functions between the two OARG-related clusters. Cluster A was primarily enriched in DNA receptor and cell cycle-related pathways and keratinization-related biological processes, and was associated with keratinocyte membrane mantle and keratin fibers (Figure 4C–F). Analysis of the tumor mutation burden showed that cluster A had a higher tumor mutation burden (Figure 4G, *p* < 0.001). Stemness index analysis indicated that mRNAsi, EREG-mRNAsi, and EREG-mDNAsi of cluster A were closer to 1, suggesting that NSCLC patients in cluster A had a higher similarity with stem cells and a higher degree of malignancy (Figure 4H).

### 2.5. Construction of OARG-Related Risk Scoring Model

To explore the clinical value of OARGs, risk score models were constructed using TCGA data and GSE42127 data. The first risk score model was constructed by Lasso–Cox analysis, and the lambda.min was 3 (Figure 5A–F). Then, the second risk score model was constructed using Random Survival Forest (RSF) (Figure 5G–O). The optimized nodesize = 20 and mtry = 8 were obtained in the first grid search, and the out-of-bag error rate was 34.66%. Three variables, SLC2A1, LDHA and PLAU, were selected by the minimum depth method, and the optimized Random Survival Forest was established by a second grid search. The second grid search results in nodesize = 4, mtry = 1, and its out-of-bag error rate is 34.19%. Finally, the gradient boosting machine (GBM) was used to construct the third risk scoring model (Figure 5P–S). Through continuous debugging, the GBM model was finally constructed with n.trees = 3000, learning rate shrinkage = 0.001, interact.depth = 2, n.minobsinnode = 5 and nine-fold crossover operation. NSCLC patients were divided into a high-risk group and low-risk group according to the risk score calculated by the model. The KM curve and log-rank test demonstrated that high-risk patients had a higher risk of death in both the training set and the test set, indicating that the risk scoring models established were effective. The time-dependent ROC curve showed that the RSF model had the best prediction performance, with AUC values of 0.654, 0.661, and 0.608 at 1 year, 3 years, and 5 years in the training set, and 0.697, 0.618, and 0.574 in the test set, respectively.

### 2.6. Establishment of Prognostic Nomogram for NSCLC Patients

Multivariate Cox regression analysis identified age, stage and risk score as independent prognostic risk factors for NSCLC patients (Figure 6A). To incorporate clinicopathological factors into the prediction model, we combined the risk score model with clinical information to construct a nomogram (Figure 6B). The calibration curve demonstrated the high validity of the nomogram (Figure 6C). The cumulative risk curve also showed a progressive increase in the risk of overall survival in NSCLC patients with higher scores on the nomogram (Figure 6D). Decision curve analysis (DCA) at 1, 3, and 5 years showed that both the nomogram and risk score were good methods for predicting the short-term and long-term survival of NSCLC patients (Figure 6E–G).

### 2.7. Relationship between Different Risk Scores and OARG-Related Characteristics, TMB, Stemness and Drug Sensitivity in NSCLC Patients

The relationship between the risk score and OARG-related clusters revealed that cluster A had a significantly higher risk score than cluster B, further confirming the association of cluster A with poor prognosis (Figure 7A, *p* < 0.001). The Sankey diagram also showed that cluster A was mainly clustered in the high-risk group (Figure 7B). The heatmap displayed that PLAU, SLC2A1 and LDHA involved in the model were highly expressed in the high-risk group. From the heatmap, we can also observe that male patients and lung squamous cell carcinoma patients were mainly clustered in the high-risk group, which once again validated that the high expression of OARGs in male patients and lung squamous cell carcinoma patients was associated with poor prognosis (Figure 7C). The analysis of tumor mutation burden showed that the TMB of the high-risk group was significantly higher than that of the low-risk group (Figure 7D, *p* < 0.001), and the risk score was positively correlated with TMB (Figure 7E, R = 0.14, *p* < 0.001). The stemness index showed that the mRNAsi, EREG-mRNAsi, and EREG-mDNAsi of the high-risk group were closer to 1 (Figure 7F), indicating that the cancer cells in the high-risk group were more malignant. Finally, the potential sensitivity of the high-risk and low-risk groups to clinical agents was explored using the R software 4.3.0 “oncoPredict” package (Appendix A).

### 2.8. Relationship between Risk Score and Immune Cell Infiltration and TME

M0 macrophages accounted for a large proportion of immune cell components in NSCLC patients, and their proportion was higher in the high-risk group than the low-risk group (Figure 8A, *p* < 0.001). Correlation analysis found a positive relationship between the proportion of M0 macrophages and the risk score (Figure 8B, R = 0.27, *p* < 0.001), and correlations between other immune cells and risk scores are presented in the Appendix A. These findings suggest that M0-type macrophages may be an important cause of poor prognosis in NSCLC patients with high OARG expression. Furthermore, correlations among immune cells in NSCLC patients provided insights into the composition of the immune microenvironment of specific types of tumors (Figure 8C). The correlation between hub OARGs in the model and immune cells was also investigated. We found that SLC2A1, LDHA, and riskScore were strongly correlated with immune cell levels in NSCLC patients. (Figure 8D), which indicated that the risk score model we constructed can be used as a tool to predict the effect of immunotherapy. Additionally, the ESTIMATE algorithm was used to obtain the stromal score, immune score, and ESTIMATE score of the high-risk and low-risk groups (Figure 8E). The low-risk group had a higher stromal score, immune score, and ESTIMATE score, indicating that the low-risk group had lower tumor purity and better prognosis.

### 2.9. Immune Checkpoint Expression, Immune Cell Function Score and IPS Score in High- and Low-Risk Groups

The expression levels of common immune checkpoint genes were compared between the high- and low-risk groups, revealing higher expression levels in the low-risk group, including PD-1 (PDCD1), PD-L1 (CD274), CTLA4, CD80, and others (Figure 9A). This suggests that immunotherapy in the low-risk group would have a better effect. In addition, the levels of most immune functions were higher in the low-risk group (Figure 9B), such as aDCs, APC_co_stimulation, CD8+T cells, and HLA (*p* < 0.001), which further validated the better response to immunotherapy in the low-risk group. The IPS score of low-risk patients who received different types of immune checkpoint blockade was significantly higher than that of high-risk patients (Figure 9C–F), confirming the results of immune checkpoints and immune function.

### 2.10. Correlation between OARGs Involved in Risk Score Model and Tumor Immune Microenvironment (TME)

We used the single-cell dataset NSCLC_GSE131907 from the TISCH database to analyze the expression of three OARGs involved in the risk score model in the TME. In the GSE131907 dataset, 12 major cell clusters are shown, along with their distributions and numbers (Figure 10A,B). SLC2A1 was mainly expressed on monocytes/macrophages and epithelial cells, compared to lower expression in fibroblasts (Figure 10C,D). LDHA showed relatively high expression in various cells, especially epithelial cells and monocytes/macrophages (Figure 10E,F). PLAU expression was high in epithelial cells and monocytes/macrophages, but its expression was also high in fibroblasts and DC cells (Figure 10G,H). These results suggest that epithelial cells and monocytes/macrophages play a key role in oxidative stress-related anoikis.

### 2.11. Competitive Endogenous RNA (ceRNA) Network Analysis of LDHA

Initially, 39 mRNA-miRNA relationship pairs were predicted from the miRWalK database (Appendix A). Subsequently, 236 miRNAs differentially expressed in NSCLC were identified by GSE36681 (Appendix A). By intersecting these two sets of miRNAs, we obtained five miRNAs that were not only associated with the hub genes (SLC2A1, LDHA), but also relevant to NSCLC (Appendix A). Next, we utilized the ENCORI database to predict the miRNA–lncRNA relationship pairs using these 5 miRNAs, and 96 pairs of interactions were predicted, consisting of 1 mRNA (LDHA), 3 miRNAs (has-miR-338-3p, has-miR-330-5p and has-miR-34c-5p), and 96 lncRNAs (Appendix A). Finally, we constructed a complex ceRNA network based on these relationship pairs (Figure 11, Appendix A).

### 2.12. Validation of Expression Levels of SLC2A1, LDHA, PLAU

The IHC results from the HPA database showed that SLC2A1, LDHA and PLAU were all highly expressed in NSCLC tissues at the protein level (Figure 12A–C). Data from GSE101929 and GSE74706 also showed that SLC2A1, LDHA and PLAU were differentially expressed between tumor and normal tissues, and were highly expressed in tumor tissues (Figure 12D–I). These results further demonstrated the accuracy and potential value of the biomarkers screened in this study.

## 3. Discussion

Anoikis is a special type of programmed cell death that occurs when normal epithelial cells lose their connection with the extracellular matrix. This process helps eliminate misaligned or displaced cells, preventing the possibility of malignant tissue proliferation and influencing cancer development [35]. The study found that anoikis in fact can be adjusted by oxidative stress. AGP has been shown to activate Nrf2, reduce the reactive oxygen species (ROS) level, and mitigate mitochondrial dysfunction and anoikis in mice colon epithelial cells [24]. Another gene, Succinyl coa ligase ADP form subunits beta (SUCLA2), can promote the formation of the stress particles to further promote the antioxidant enzyme protein translation, and thus reduce oxidative stress and endow cancer cells with resistance to anoikis [36]. In addition, polygodial (PG), a natural sesquiterpene compound isolated from water pepper, Dorrigo pepper and mountain pepper, was found to exert anticancer effects by promoting the production of ROS and inducing apoptosis in castration-resistant prostate cancer cells [37]. Existing studies have developed risk models focusing on either oxidative stress or anoikis alone to predict survival and immune status in various types of malignant tumors [38,39,40,41]. However, few risk models combine both phenotypes.

In this study, we identified 47 differentially expressed genes related to oxidative stress and anoikis in NSCLC. These genes mainly regulate cells in response to chemical stress and epidermal cell infiltration through the PI3K-Akt signaling pathway and MicroRNAs in cancer. Among these 47 OARGs, 10 were closely associated with poor prognosis in NSCLC, with BRCA1 having the highest somatic mutation frequency. Using these 10 prognostic genes, we identified two OARG-related clusters. Cluster A, which has higher somatic mutation frequency and a stemness index closer to 1, has a worse prognosis and lower immune infiltration level, including activated B cells, activated CD8T cells, activated dendritic cells, eosinophils and macrophages. Immune cell infiltration has been shown to impact the response to immunotherapy and the expression of immune checkpoints [42,43,44,45]. For instance, CD8T cells, as a subset of PD-1 T cells, have a higher production of interferon-γ and granzyme B than peripheral blood monocytes in lung cancer [46], while a higher proportion of dendritic cells is associated with better overall survival of lung cancer [47]. 

Three risk score models were constructed based on these 10 prognostic OARGs, with the Random Survival Forest model performing the best. The risk score was calculated by incorporating three hub genes: SLC2A1, LDHA, and PLAU. SLC2A1 and LDHA, important regulators of the glycolytic pathway, were found to be up-regulated in precancerous lesions of lung squamous cell carcinoma, and further up-regulated in invasive carcinoma of lung squamous cell carcinoma, indicating that they may be driver genes of lung cancer [48]. Inhibition of SLC2A1 can lead to apoptosis due to insufficient glucose transport, while inhibition of LDHA increases oxidative phosphorylation in mitochondria, resulting in elevated ROS levels and cytotoxic effects on cancer cells. [49]. In addition, the upregulation of SLC2A1 resulted in the overexpression of NRF2 [50], a central regulator of ROS transcriptional responses [51], suggesting that SLC2A1 could affect anoikis through oxidative stress. When the activity of LDHA is restrained, cancer cells become sensitive to anoikis and the increased ROS [52]. And PLAU in lung cancer is closely related to the tumor suppressor gene TP53 mutations, and can prevent the occurrence of anoikis [53,54]. According to the risk score, NSCLC patients were divided into high- and low-risk groups, with the prognosis significantly better in the low-risk group. In addition, the tumor mutation burden and stemness index of the high-risk group were higher than those of the low-risk group, which also indicated that the patients in the high-risk group had a higher degree of malignancy. Multivariable Cox regression analysis confirmed the significant predictive benefits of using the risk score as an independent prognostic factor in NSCLC patients. The constructed nomogram combining the risk score and other clinical characteristics demonstrated high predictive efficiency, as confirmed by the calibration and decision curve analyses.

The tumor microenvironment (TME) is composed of various cellular components (immune cells, stromal cells, fibroblasts, etc.) and extracellular components (cytokines, growth factors, extracellular matrix, etc.) [55]. The TME plays an important role in the development of cancer, metastasis and tumor cells to evade the host immune [56]. We found that there was a significant difference in the proportion of the tumor microenvironment in NSCLC patients between the high- and low-risk groups. M0 macrophages, the predominant immune cell population in NSCLC, were more abundant in the high-risk group, and their proportion increased with the risk score. This suggests that M0 macrophages may contribute to poor prognosis in patients with a high expression of OARGs. In addition, SLC2A1 and LDHA were found to be highly correlated with the degree of immune cell infiltration, particularly M0 macrophages and activated mast cells, indicating the involvement of the SLC2A1/LDHA/M0 macrophage axis in mediating oxidative stress-related anoikis. The results of the single-cell dataset GSE131907 further confirmed that SLC2A1 and LDHA were closely related to monocytes/macrophages. In this study, the ESTIMATE scores of the low-risk group were significantly higher than those of the high-risk group, indicating that the low-risk group had lower tumor purity and better response to immunotherapy. To further validate our findings, we examined the status of the immune landscape in the high- and low-risk groups. Immune checkpoint expression, immune function scores, and IPS scores were also higher in the low-risk group, which further illustrated the low-risk group had better response to immunotherapy. Drug sensitivity analysis showed that there was a large difference in drug sensitivity between the high- and low-risk groups, indicating that correct calculation of risk scores is very important for guiding the use of chemotherapy drugs.

The study had shown that miRNAs and lncRNAs play an important role in regulating genes and biological processes in cancer [57]. In the ceRNA network, lncRNAs compete with miRNAs in the regulation of target genes and participate in the occurrence and development of tumors [58]. To comprehensively understand the roles and underlying mechanisms of the three hub genes in oxidative stress-related anoikis in NSCLC patients, a ceRNA network centered on LDHA was constructed. This suggests that LDHA may be the most critical gene involved in oxidative stress-related anoikis in NSCLC among the three hub genes. 

Although we comprehensively explored the characteristics of OARGs in NSCLC and constructed three OARGs risk scoring models, there are still some limitations in this study. Firstly, the amount of data in this study is limited, and larger sample sizes are needed to validate the prediction model. Second, our analyses and conclusions are based on retrospectively collecting tumor samples in public databases, which may lead to inherent case selection bias. Finally, our results lack experimental verification.

## 4. Materials and Methods

### 4.1. Data Acquisition

RNA-seq transcriptome data, micRNA data and clinical information of NSCLC with complete overall survival (OS) and survival status were downloaded from the Cancer Genome Atlas (TCGA, https://portal.gdc.cancer.gov/, accessed on 4 March 2023) (TCGA-LUAD and TGGA-LUSC) and Gene Expression Omnibus (GEO, https://www.ncbi.nlm.nih.gov/geo/, accessed on 8 March 2023) (GSE42127 and GSE36681). Somatic mutation count and copy number variation (CNV) data were also downloaded from the TCGA database. A total of 560 genes related to anoikis were downloaded from the GeneCard [59] (https://www.genecards.org/, accessed on 6 March 2023) database and Harmonizome portal (https://maayanlab.cloud/Harmonizome/, accessed on 6 March 2023) [60], and 1000 genes related to oxidative stress were downloaded from GeneCard database. In total, 185 oxidative stress-related anoikis genes (OARGs) were obtained by intersection of the two gene lists. Stemness indexes (mRNAsi, EREG-mRNAsi, mDNAsi, EREG-mDNAsi, DMPsi and ENHsi) for NSCLC were obtained from the study of *Tathiane M. Malta* et al. [61]. We used the sva R package for standardization and log2 conversion expression profile data. The ComBat algorithm was used to remove batch effects when integrating TCGA and GEO data. Finally, 47 differentially expressed genes (DEGs) were identified in the TCGA cohort by using the “limma” [62] package of R software 4.3.0.

### 4.2. Related Features of OARGs

Firstly, through Gene Ontology (GO) and Kyoto Encyclopedia of Genes and Genomes (KEGG) analysis, the related biological functions and pathways involved in 47 OARGs with differences in tumor tissues and adjacent normal tissues were explored. By merging the GEO dataset, we performed univariate Cox regression analysis on 47 OARG-related genes and found 10 OARGs associated with prognosis. The Z-score for the univariate Cox regression analysis was calculated with the Wald test, with a threshold of α = 0.05 and a *p* value of less than 0.05 considered to indicate statistical significance. Next, we investigated the interaction relationships among these 10 OARGs with prognostic value. Meanwhile, the somatic mutation rate, genetic loci and CNV of these 10 genes were analyzed.

### 4.3. Consistency Clustering to Identify OARG Clusters of NSCLC

A consensus clustering method was used to classify patients into different OARG clusters based on the expression levels of the 10 prognostic OARGs. The k-means method was used to identify the clusters with the lowest inter-group correlation and the highest intra-group correlation, and the reliability of the clustering was verified by principal component analysis (PCA) using R packages “ggplot2” and “limma”. Next, the KM method and logrank test combined with the survival and survminer R package were used to analyze the OS of different clusters of patients. Then, the clinical characteristics and differentially expressed genes (DEGs) of different OARG clusters were analyzed. Subsequently, gene set variation analysis (GSVA) and gene set enrichment analysis (GSEA) were used to evaluate the activity changes in the pathways or functions in gene sets, and single-sample gene set enrichment analysis (ssGSEA) was used to explore the infiltration of immune cells in different clusters. Finally, combined with the stemness index and tumor mutation burden (TMB) of non-small cell lung cancer, the Wilcoxon test was used to analyze the stemness characteristics and tumor mutation burden of different clusters. Among them, TMB data were calculated from NSCLC somatic mutation data and CNV data through the “matfools” package of R software 4.3.0 and the tmb function.

### 4.4. Construction of Risk Score Model

TCGA data and GSE42127 data were integrated to construct a risk score model. Using the “survival”, “survminer” and “glmnet” R packages, for 10 prognosis-related OARGs, we carried out least absolute shrinkage and selection operator (LASSO) regression analysis and multivariate Cox regression analysis to determine the center gene and construct the LASSO + Cox risk score model. Then, the Random Survival Forest (RSF) risk scoring model was constructed by using the R packages “randomForestSRC”, “caret” and “survival”, and the grid search method was used to find the optimal parameters and the minimum depth method was used to screen the variables. Finally, the Gradient Boosting Machine (GBM) risk scoring model was constructed by using the “gbm”, “limma” and “survival” packages, and the model was optimized by using the nine-fold cross validation method. Kaplan–Meier (KM) survival curve and time-dependent receiver operating characteristic (ROC) curve analysis were used to evaluate the predictive ability of the model. The model with the best predictive ability was selected to calculate the risk score. The risk score was calculated based on the expression level of the gene and the corresponding coefficient calculated by the model. Next, patients were divided into high-risk and low-risk groups based on the risk score for subsequent analysis.

### 4.5. Construction and Evaluation of Nomogram for NSCLC Patients

Multivariate Cox regression analysis was used to analyze the clinicopathological characteristics and risk scores of NSCLC patients to determine the independent prognostic characteristics of patients. Subsequently, the clinicopathological characteristics and risk scores of the patients were combined, and relevant R packages such as “rms” were used to construct a nomogram to further demonstrate the role of these independent prognostic factors. The calibration chart was used for internal validation to verify the accuracy, the cumulative risk curve was used to further verify the reliability of the nomogram, and the decision curve analysis (DCA) was used to evaluate the net clinical benefit.

### 4.6. Relationship between Risk Score and OARG Clusters, Clinical Features, Gene Expression, TMB, and Stemness Characteristics

The Wilcoxon test was used to analyze the differences of OARG subtypes, TMB and stemness characteristics under different risk scores, and Spearman rank correlation was used to analyze the correlation between risk scores and TMB. The “pheatmap” package was used to visualize the expression of genes involved in scoring and the distribution of clinical characteristics in different risk groups.

### 4.7. Immune Landscape Analysis with Different Risk Scores

The CIBERSORT algorithm was used to quantify the infiltration of immune cells in NSCLC tissues, and the Spearman method was used to analyze the correlation between the risk score and the abundance of infiltrating immune cells. Subsequently, the relationship between signature genes and immune cells was also analyzed. Finally, the estimate algorithm was used to compare the differences in the tumor microenvironment between the high-risk group and the low-risk group, including the stromal score, immune score and ESTIMATE scores. Next, the Wilcoxon test was used to compare the high- and low-risk group at some of the common immune checkpoint gene expression levels. The ssGSEA algorithm was used to calculate the immune function scores to assess the differences in immune-related functions between the high- and low-risk groups. IPS data were downloaded from The Cancer Immunome Atlas (TCIA, https://www.tcia.at/home, accessed on 7 July 2023) database. IPS was used to predict patient response to various types of immune checkpoint inhibitor (ICI) therapy, including PD-1/PDL1/PD-L2, CTLA-4, CTLA-4 and PD-1/PD-L1/PD-L2 blockade.

### 4.8. High- and Low-Risk Group Differences in Drug Sensitivity

Firstly, drug-related response information and the expression matrix were collected from the Genomics of Drug Sensitivity in Cancer (GDSC, accessed on 7 July 2023, https://www.cancerrxgene.org/). The R software 4.3.0 “oncoPredict” package was used to predict the drug sensitivity in NSCLC, and the Wilcoxon test was used to compare the differences in the sensitivity of different therapeutic drugs between high- and low-risk groups.

### 4.9. Hub Genes and the Relationship between Tumor Microenvironment

The Tumor Immune Single-cell Hub (TISCH, http://tisch.comp-genomics.org/, accessed on 21 August 2023) is a large online database of single-cell RNA-seq focused on tumor microenvironments (TME) [63]. This database was used to systematically investigate TME heterogeneity and cell types in various data sets. The aim of this study is to analyze the relationship between hub OARGs and tumor immune microenvironments in NSCLC.

### 4.10. Competitive Endogenous RNA (ceRNA) Network Analysis of LDHA

The miRWalk [64] (http://mirwalk.umm.uni-heidelberg.de/, accessed on 28 August 2023) database is used to predict mRNA–miRNA relationship pairs while considering miRDB filtering, and 39 miRNAs related to three hub genes, SLC2A1, LDHA and PLAU, were screened. When the adjusted P value was less than 0.05 and |logFC| greater than 2, we obtained 236 differentially expressed miRNAs of GSE36681. The intersection of the two different types of miRNAs yielded five miRNAs that were both associated with two hub genes (SLC2A1, LDHA) and NSCLC. Next, we used the five miRNAs in ENCORI (https://rnasysu.com/encori/, accessed on 29 August 2023) database [65] to predict the miRNA–lncRNA relationship pairs. Finally, 96 pairs of interactions were predicted, which contained 1 mRNA, 3 miRNAs, and 96 lncRNAs. Finally, the ceRNA network containing these relationships was drawn by the R software 4.3.0 package “igraph”.

### 4.11. Validation of Expression Levels of SLC2A1, LDHA, PLAU

To further verify the protein expression levels of SLC2A1, LDHA and PLAU in non-small cell lung cancer and normal tissue, we downloaded immunohistochemistry (IHC) data from The Human Protein Atlas (HPA, https://www.proteinatlas.org/, accessed on 24 August 2023). RNA-seq data from GSE101929 and GSE74706 were also downloaded, which again demonstrated that SLC2A1, LDHA and PLAU were differentially expressed between normal and tumor tissues at the transcriptome level.

### 4.12. Statistical Analysis

Wilcoxon tests were used for comparisons between two groups, while Kruskal–Wallis (K-W) tests were used for comparisons among multiple groups. The Kaplan–Meier (K-M) method is employed to describe the survival curves of different groups, while the log-rank test was employed to compare different survival curves. Spearman rank correlation analysis was conducted to calculate the correlation between variables. The chi-square test was adopted for examining the mutation frequencies. All statistical analyses were performed using R software version 4.3.0. *p* < 0.05 was considered statistically significant.

## 5. Conclusions

In this study, we developed a novel prognostic signature based on oxidative stress-related anoikis genes (OARGs) and constructed a machine learning risk scoring model and a nomogram based on this signature. Our findings demonstrate the robust predictive ability of this signature in determining the prognosis, immune landscape, and treatment response of non-small cell lung cancer (NSCLC) patients. Consequently, this signature holds great potential in guiding clinicians towards personalized treatment strategies in clinical practice. Because anoikis in our study is a key mechanism of tumor metastasis, risk score models identified high-risk patients as more likely to develop metastasis. Therefore, physicians should closely monitor the condition and treatment response of high-risk patients and follow up regularly to prevent complications and recurrence. High-risk patients are also associated with poor prognosis. Doctors should perform radiotherapy and chemotherapy as early as possible to improve the prognosis of patients at high risk. In addition, the results of this study suggest that the benefit of immunotherapy is greater in low-risk patients. In addition, as oxidative stress can mediate anoikis, we can find some ways to prevent cancer metastasis by increasing the appropriate level of oxidative stress in cancer patients in the future. Finally, we also found that OARGs highly expressed in men and lung squamous cell carcinoma patients were associated with poor prognosis, so we should pay attention to men and lung squamous cell carcinoma patients with this signature high expression. In this signature, LDHA, M0 macrophages and the PI3K-AKt pathway play a crucial role in oxidative stress-related anoikis in NSCLC. In the future, studies on the relevant molecular mechanisms and prospective randomized clinical trials of this signature will be of great clinical importance.

## Figures and Tables

**Figure 1 ijms-24-16188-f001:**
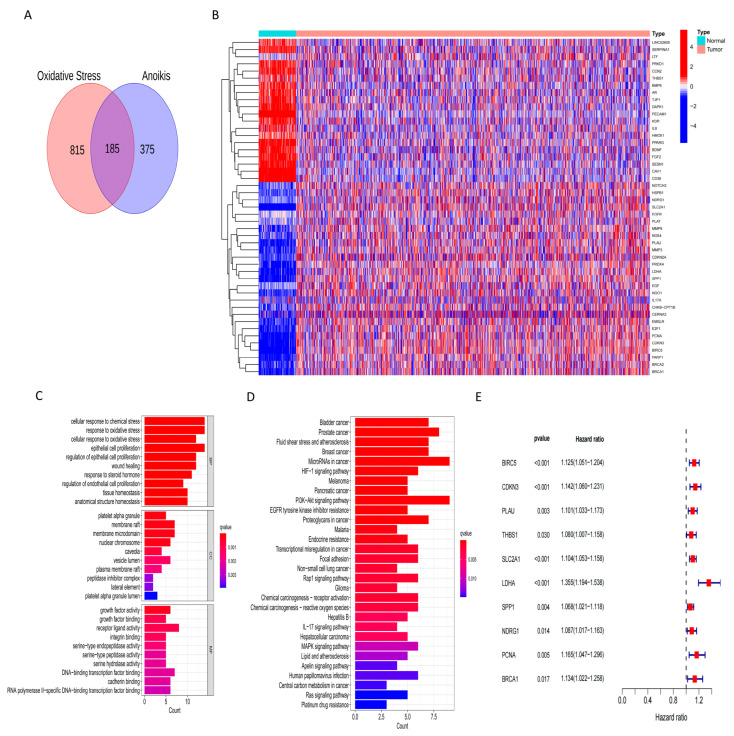
Identification of OARGs associated with prognosis. (**A**) 185 genes at the intersection of anoikis and oxidative stress. (**B**) 47 Differentially expressed OARGs in normal and tumor tissues (red indicates high expression and blue indicates low expression in the heat map). (**C,D**) Differentially expressed genes are involved in functions and pathways. (**E**) 10 prognostic-related OARGs.

**Figure 2 ijms-24-16188-f002:**
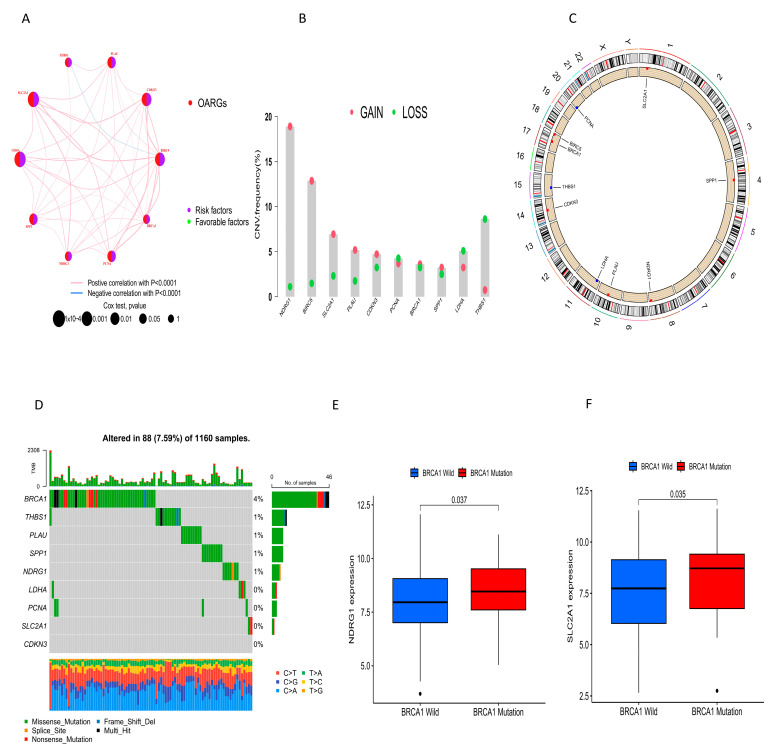
Characteristics of OARGs associated with prognosis. (**A**) Relationships between 10 prognostic-related OARGs. (**B**) CNV of prognostic-related OARGs. Green represents a decrease in the gene copy number variation and red represents an increase. (**C**) Location of CNV in prognostic-associated OARGs on human chromosomes. Blue dots represent a higher frequency of copy number loss and red represents a higher frequency of copy number gain. (**D**) Occurrence of somatic mutations in prognostic-related OARGs. (**E**,**F**) Expression differences of prognostic-related OARGs between wild-type and mutant types.

**Figure 3 ijms-24-16188-f003:**
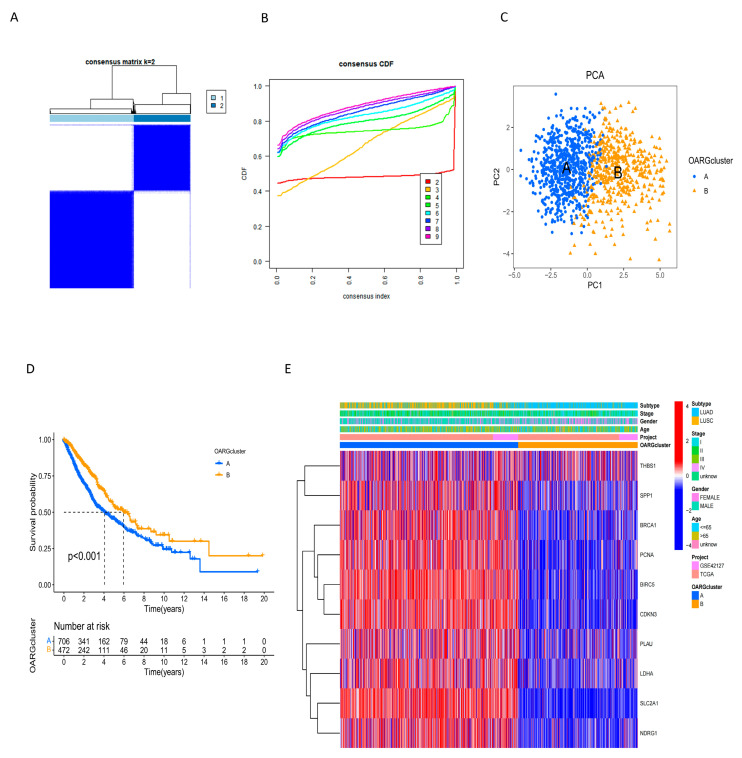
Consensus clustering of OARGs. (**A**) A total of 1193 NSCLC patients were divided into two clusters by consensus clustering. The light blue 1 area represents cluster A, and the dark blue 2 area is cluster B. (**B**) The consistent cumulative distribution function (CDF) plot shows that the CDF reaches its maximum value when k is set to 2 (k indicates several clusters). (**C**) PCA showed the distinction between the two clusters. (**D**) The KM curve revealed a significant difference in survival time between the two clusters (*p* < 0.001). (**E**) The relationship between OARG clusters and clinical features and OARG expression in NSCLC patients. (Red represents expression values above the mean and is denoted as positive, while blue represents expression values below the mean and is denoted as negative. The darker the color, the more different it is from the mean).

**Figure 4 ijms-24-16188-f004:**
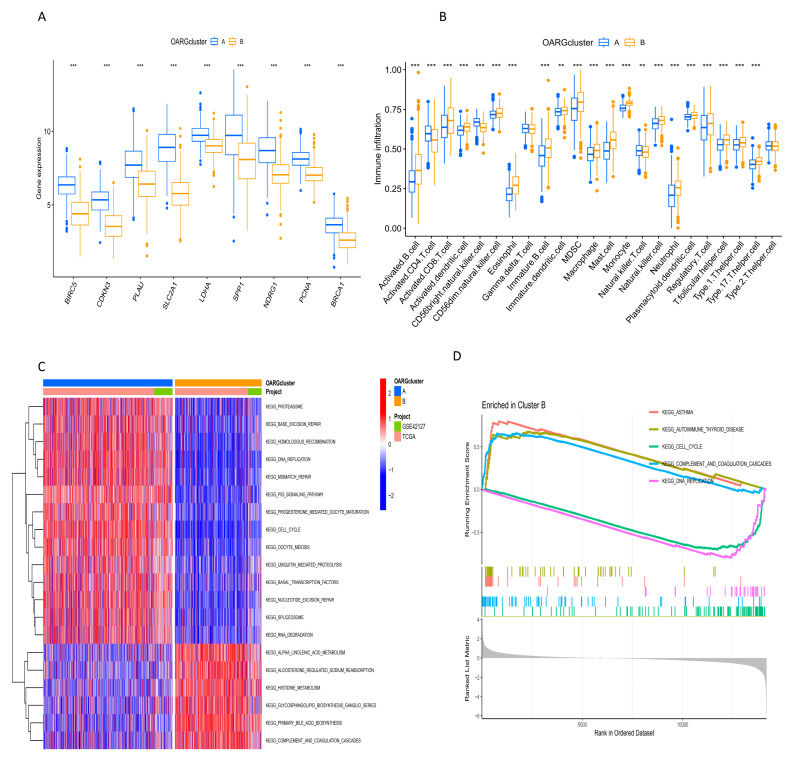
Analysis of related characteristics of OARG clusters. (**A**) Expression of OARGs in different clusters. (**B**) The ssGSEA analysis showed the infiltration of immune cells in different clusters. (**C**–**F**) GSVA analysis and GSEA analysis revealed pathways and functions that were enriched in the two clusters, respectively. (**G**) Tumor-mutation burden in the two clusters. (**H**) The situation of different stemness indices in the two subpopulations. * *p* < 0.05, ** *p* < 0.01, *** *p* < 0.001.

**Figure 5 ijms-24-16188-f005:**
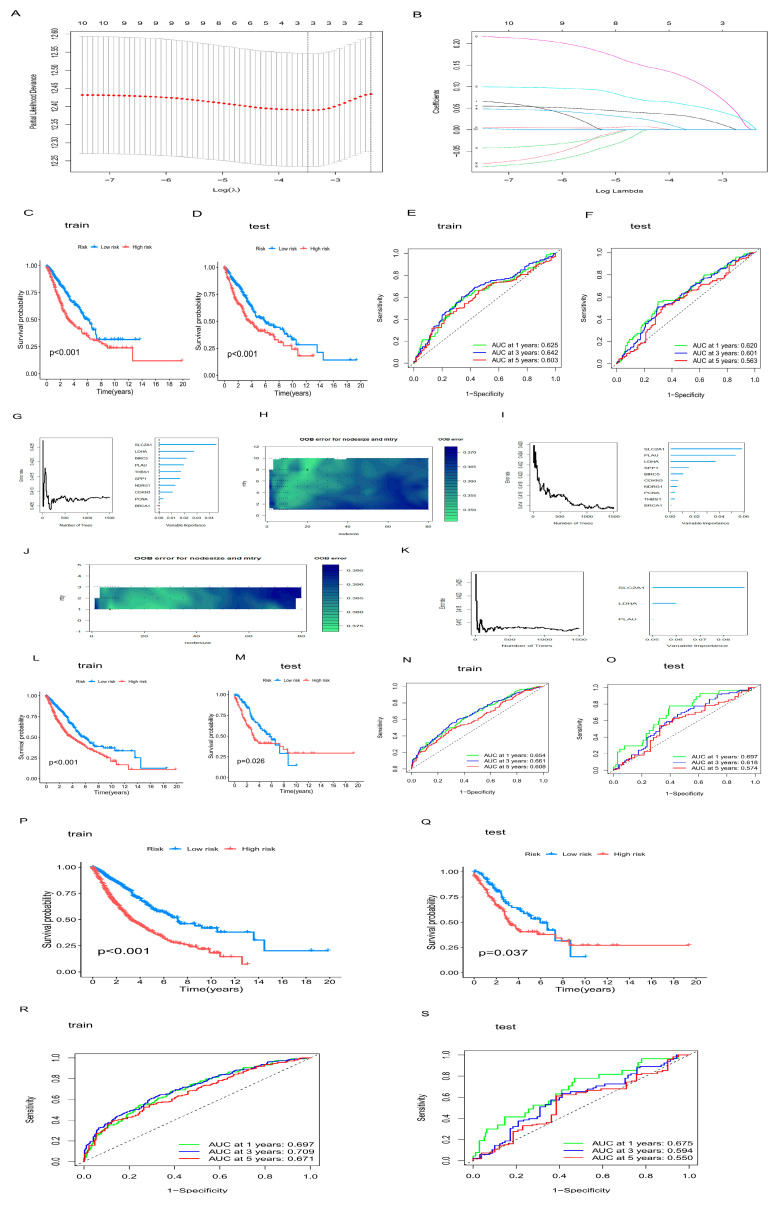
Construction of OARG-related risk scoring model. (**A**–**F**) Lasso+Cox model. (**A**): Lasso regression of lambda graph. (**B**): Coefficient plot for Lasso regression. The numbers on the upper abscissions represent the number of nonzero coefficients in the model; The ordinate represents the magnitude of the gene coefficients; The lower abscissa represents the normalized coefficient vector; The numbers on each line represent the number of the different genes. (**C**,**D**): KM curves for the training and test sets. (**E**,**F**): ROC curves for the time dependence of the training and test sets. (**G**–**O**) Random Survival Forest model. (**G**): The model was built for the first time. (**H**): The model was constructed by a grid search to find the optimal parameters. (**I**): The optimized model was established. (**J**): Grid search after filtering variables by the minimum depth method. (**K**): The model was built after screening the variables and optimizing. (**L**,**M**): KM curves for the training and test sets. (**N**,**O**): ROC curves for the time dependence of the training and test sets. (**P**–**S**) GBM model. (**P**,**Q**): KM curves for the training and test sets. (**R**,**S**): ROC curves for the time dependence of the training and test sets.

**Figure 6 ijms-24-16188-f006:**
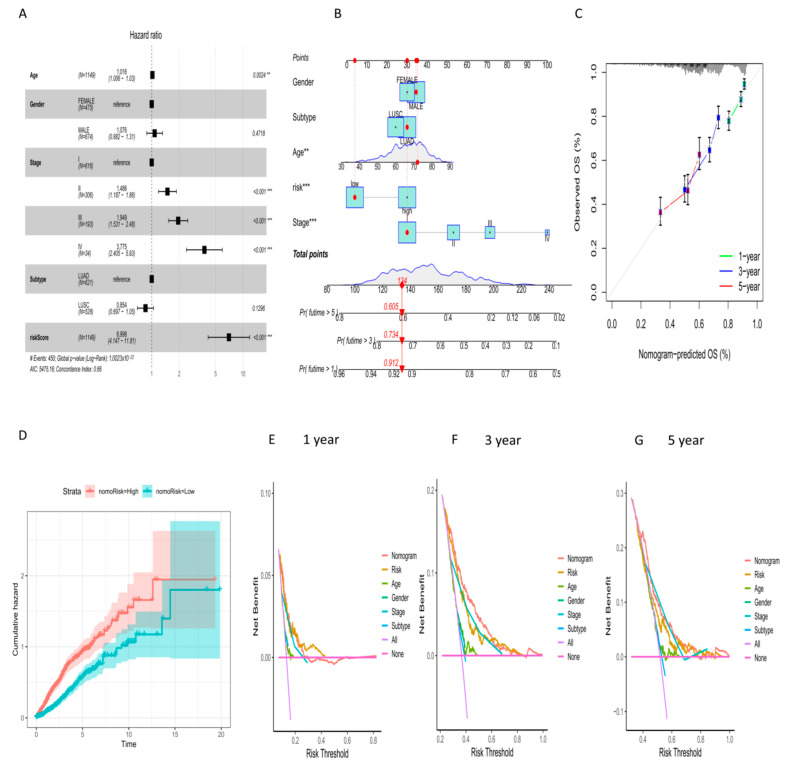
Establishment of nomogram for NSCLC patients. (**A**) Multivariate Cox regression analysis in NSCLC patients. (**B**) Nomogram constructed by risk score and other clinicopathological factors. (**C**) Calibration plot showed the differences between nomogram-predicted survival rates and actual survival rates. (**D**) Cumulative hazard curve showed the probability of survival over time progression based on different scores. (**E**–**G**) DCA curves of the nomogram at 1, 3 and 5 years. Decision making based on risk scores had higher clinical benefits in the first year, while both the third and fifth years had higher benefits based on the nomogram. ** *p* < 0.01, *** *p* < 0.001.

**Figure 7 ijms-24-16188-f007:**
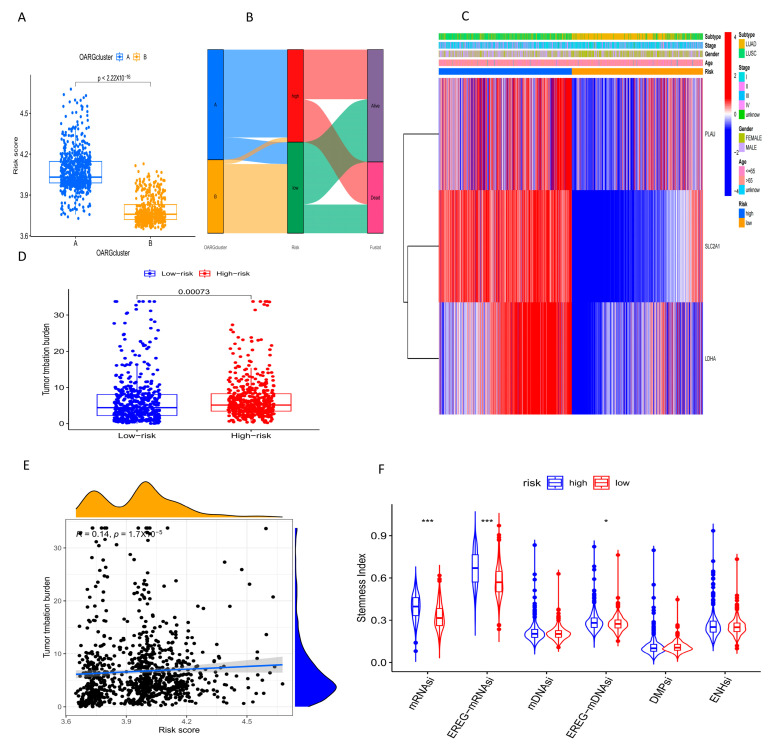
Relationship between different risk scores and OARG-related characteristics. (**A**) Differences in risk scores in subgroups A, B. (**B**) The Sankey plot shows that group A is mainly distributed in the high-risk group. (**C**) Hub genes with risk scores and clinical features of patients with NSCLC. (Red indicates high expression and blue indicates low expression). (**D**) Differences in TMB between the high- and low-risk groups. (**E**) Spearman rank correlation analysis found a positive correlation between TMB and risk scores. (**F**) Differences in stemness index between high- and low-risk groups. * *p* < 0.05,*** *p* < 0.001.

**Figure 8 ijms-24-16188-f008:**
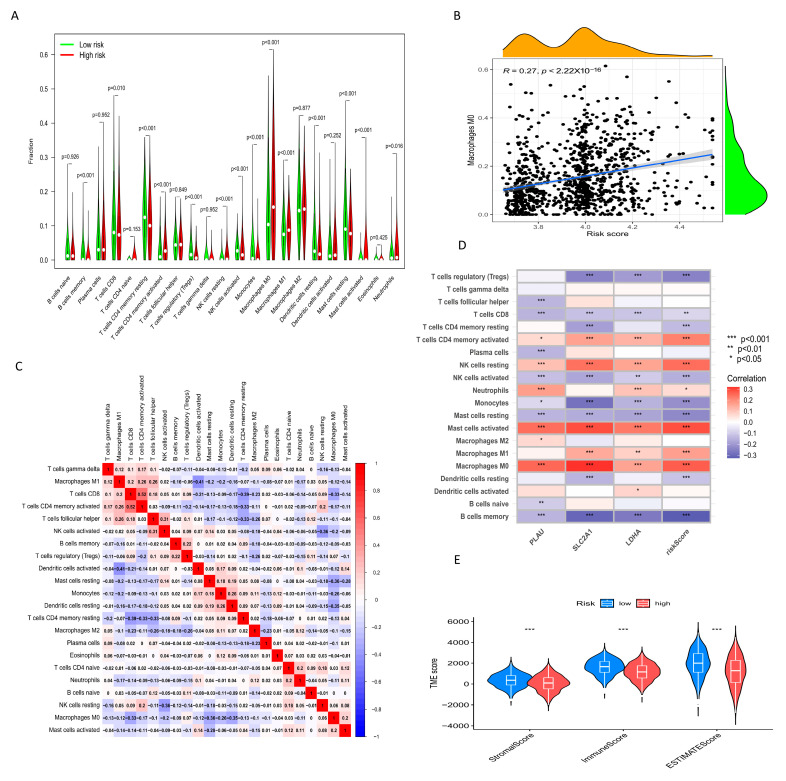
Relationship between risk score and immune cell infiltration and TME. (**A**) Differences in the component of different immune cells between the high- and low-risk groups. (**B**) Correlation analysis indicated that M0 macrophages and risk scores were positively correlated. (**C**) Correlation between immune cells. (**D**) Correlation between immune cells and 3 hub OARGs. (**E**) StromalScore, ImmuneScore and ESTIMATEScore in high-risk group and low-risk group. * *p* < 0.05, ** *p* < 0.01, *** *p* < 0.001.

**Figure 9 ijms-24-16188-f009:**
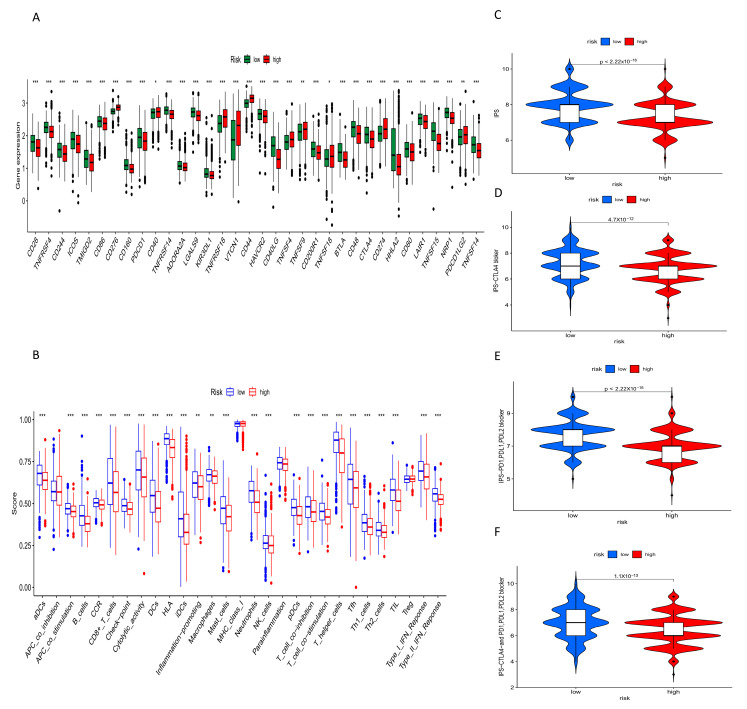
Immune checkpoint expression, immune cell function score and IPS score in high- and low-risk groups. (**A**) Differences in immune checkpoint expression between high-risk and low-risk groups. (**B**) Differences in immune function between high- and low-risk groups. (**C**–**F**) Differences in IPS scores between high- and low-risk groups. * *p* < 0.05, ** *p* < 0.01, *** *p* < 0.001.

**Figure 10 ijms-24-16188-f010:**
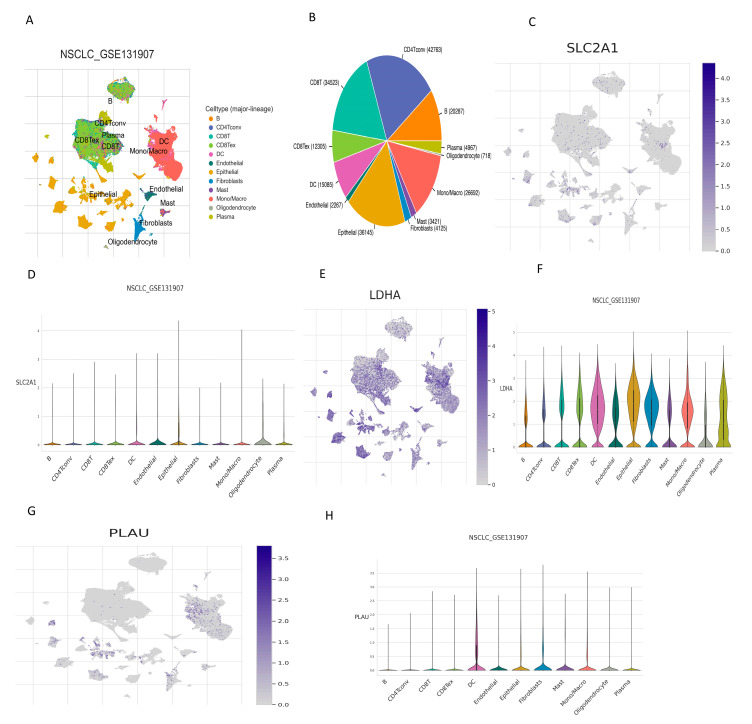
Correlation between OARGs involved in risk score model and tumor immune microenvironment. (**A**,**B**) Annotation of 12 major cell clusters in GSE131907 and the percentage of each cell clusters (**C**,**D**) Percentage and expression of SLC2A1. (**E**,**F**) Percentage and expression of LDHA. (**G**,**H**) Percentage and expression of PLAU.

**Figure 11 ijms-24-16188-f011:**
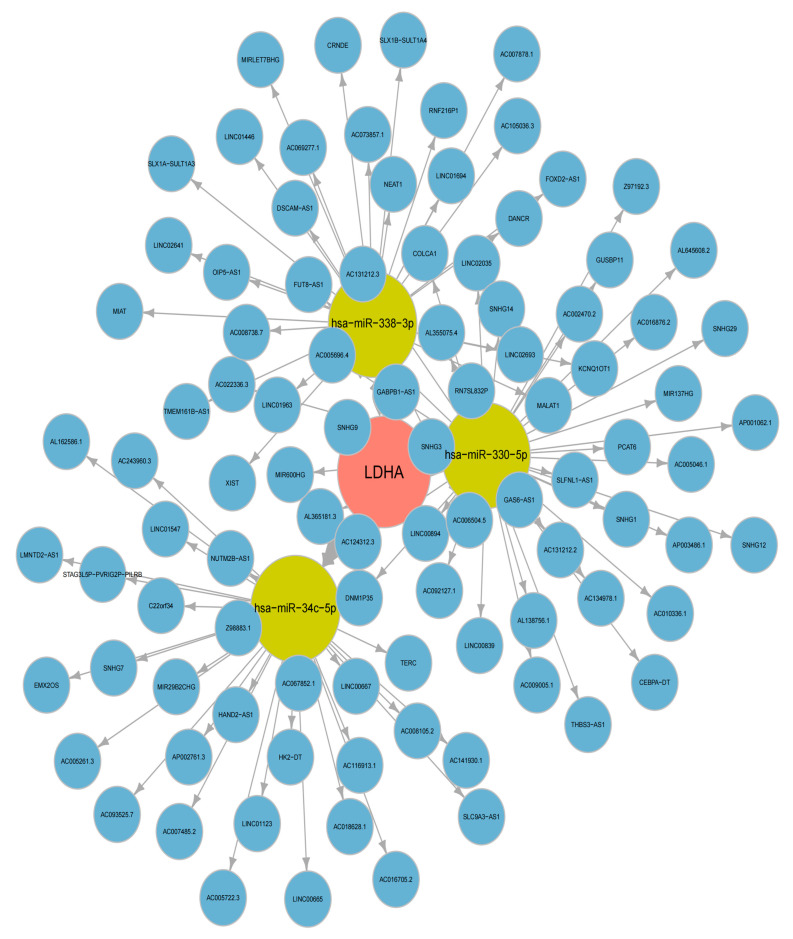
Competitive endogenous RNA (ceRNA) network analysis of LDHA.

**Figure 12 ijms-24-16188-f012:**
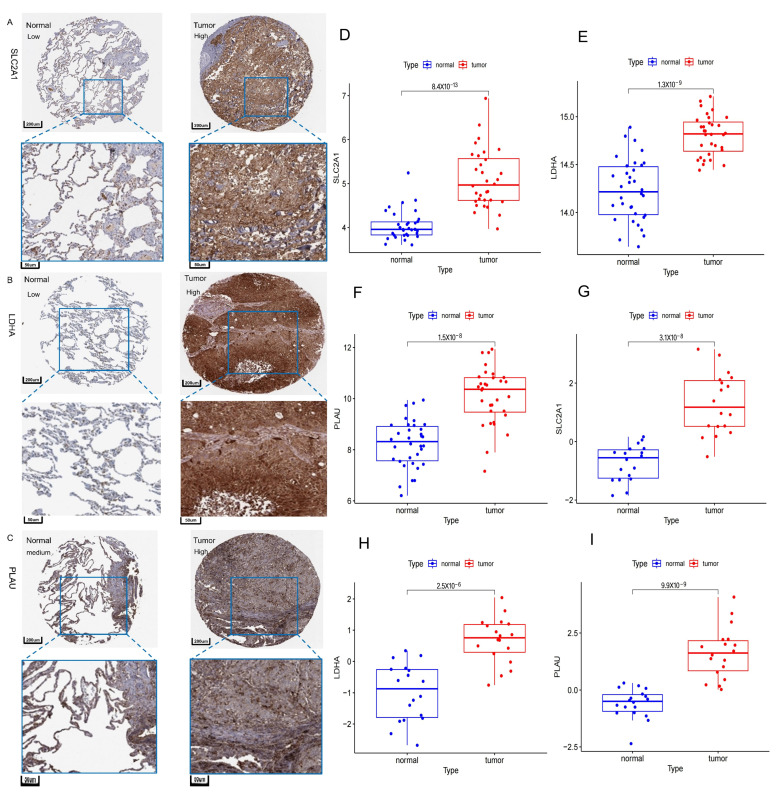
Validation of expression levels of SLC2A1, LDHA, PLAU. (**A**–**C**) Immunohistochemical results for SLC2A1, LDHA, and PLAU from HPA database. (**D**–**F**) Expression levels of SLC2A1, LDHA, and PLAU in the GSE101929 dataset. (**G**–**I**) Expression levels of SLC2A1, LDHA, and PLAU in the GSE74706 dataset.

## Data Availability

The datasets used and analyzed during the current study are available from the corresponding author on reasonable request. The data that support our results from the current study are available on The Cancer Genome Atlas (TCGA) and Gene Expression Omnibus (GEO) websites.

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
