# Peer review of "Identification of a Novel Oxidative Stress- and Anoikis-Related Prognostic Signature and Its Immune Landscape Analysis in Non-Small Cell Lung Cancer"

_ijms, 2023, doi:10.3390/ijms242216188_

Round 1

Reviewer 1 Report

Comments and Suggestions for Authors

I find the manuscript excellent. The Authors do provide a possibile prognostic tool in a field of clinical oncology where prognostic tools lack.

However, just a very little comment is to be made. I would like the Authors to write a brief comment in the manuscript of the real applicability of this tool in everyday clinical practice, since at the moment I found it hard to be routinely applicable. 

Reviewer 2 Report

Comments and Suggestions for Authors

The article „Identification of a novel oxidative stress- and anoikis-related prognostic signature and its immune landscape analysis in non-small cell lung cancer” by Zhao et al. reports the discovery of a prognostic profile bioinformatically investigating public in silico gene expression data of NSCLC patients. The authors aimed to explore the relevance of oxidative stress and anoikis associated genes and their therapeutic potential in NSCLC.

Overall, the structure of the article is reasonable; adequate methodology was utilized and data are represented in chronological analysis order. Subfigures are presented scientifically correct. Nevertheless, font sizes are often too small, annotations are missing or given details (like headers or shortages) are not explained in the text or at least in the figure legends. It is often not clear to the reader, data originating from which public projects (TCGA?, GSE?) is illustrated in the figures. The figure legends require some more detailed information.

Methods were described well.

Unfortunately, the English language is in parts difficult to understand, error-prone and requires major revision.

In addition, the text of the results section is unsatisfactory and does not give sufficient detail/explanation about the presented findings. This might be due to the enormous amount of data presented. The content of subfigures is basically listed per se, but often not interpreted or discussed. Here an example: “The boxplot showed the expression pattern of OARGs in the three clusters (Figure 4A), and the result was similar to those of the heatmap.” (p. 5, line 152-153). What is the expression pattern? Which genes are listed? Which heatmap do the authors refer to? In sum, it is hard to follow the content of the text and often no causal relations between the findings are drawn. The amount of clusters presented is highly confusing.

For scientific correctness, all gene names should be written cursively and it is recommended to add gene identifiers (RefSeq) to each supplementary table.

Finally, the authors should consider reducing the load of data presented to bring some more clarity. For instance, characteristics of OARGS could be moved to the supplementary because they are not that relevant for the argumentation of the prognostic value of OARGs.

Further detailed issues can be found below.

Detailed comments:

1.    P. 1, line 29: “to releaveled” must be “to reveal”. Please correct.

2.    P. 3, line 96-97: Genes of which data set was investigated here? Please mention the cohort name or GSE number.

3.    P. 4, Figure 2C: What does the color code “blue” and “red” of the dots representing each gene mean?

4.    P. 4, line 121-122: The description of the gene locations on chromosomes is too vague. Please outline more detailed what is shown in Figure 2C.

5.    P. 4, line 124: Please replace “Meanwhile” by “Moreover”.

6.    P. 4, line 124-126: Complicated verbalization. Basically, NDRG1 and SLC2A1 are significantly mutated. Please rephrase.

7.    P. 5, Figure 3A: Please add some annotations to the cluster: What is shown in light blue and dark blue? How many samples were analyzed (n number)? Which cluster algorithms were applied?

8.    P. 5, Figure 3B: What is presented by the nine curves? What does CDF mean? Such information should be explained in the figure legend. Please discuss the given results in the text (paragraph 2.3).

9.    P. 5, Figure 3E: What does the color code “red” and “blue” represent (p-values, expression levels) ? Please explain in the legend.

10.  P. 5, Figure 3 C: Which cluster A or B is matching the cluster 1 or 2 depicted in Figure 3A? Give more explanation in the text (paragraph 2.3).

11.  P. 13, Figure 8 A: This clustered heatmap is meaningless due to lack of details. Please remove this heatmap. Figure 8 C describes the difference of immune cell profiles in cluster A and B well enough.

12.  P. 14, Figure 9 A, B: Both box blot graphs are highly confusing. Significances cannot clearly be matched to the corresponding genes. Please increase the gap between adjacent data

13.  P. 15, Figure 10 A-C: Figure A can be omitted from figure 10 because it presents 25 clusters, while B-H focuses only on 12 clusters.

14.  P. 19, line 434: Which TCGA data set was analyzed? Please add the project ID (TCGA-LUAD?).

15.  P. 19, line 435: Wrong link. Please give the correct link to GEO database https://www.ncbi.nlm.nih.gov/geo/ .

16.  P. 19, 446: Please cite the “limma” Package of R software (PMID: 2560792).

17.  P. 20, line 452: Please give more information: statistical test, threshold, p-values of the 10 OARGs.

18.  P. 20, line 454: What is the relevance of genetic loci and CNV analyzes?

Comments on the Quality of English Language

The English language requires some moderate editing. 

Round 2

Reviewer 2 Report

Comments and Suggestions for Authors

Dear authors,

please see attached word file for the reviewer's responses.

Kind regards.

Author Response

Comments 1: [Now it is clear which genes the authors refer to. However, an interpretation of the depicted data is missing. The authors should describe the expression pattern, meaning that the given genes are significantly upregulated in cluster A compared with cluster B. Please add some explanatory text (p. 8, line 172).]

Response 1: Special thanks to you for your good comments. We have re-written this part according to your suggestion.The details are as follows. [All given genes are significantly upregulated in cluster A compared with cluster B (Figure 4A), corroborating the findings from the heatmap (Figure 3E). It also justified the conclusion that cluster A has a shorter survival period than cluster B (Figure 3D). Because the expression of these genes is a risk factor for the prognosis of NSCLC patients (Figure 1E). Moreover, these genes were all highly expressed in cluster A, indicating that cluster A was associated with poor prognosis and shorter survival.]” This change can be found on the p8, line 170-176.

Comments 2: [My question was, why the genetic loci is relevant for the transcriptomic analysis of NSCLC clusters? Your manuscript contains a ton of information; in my opinion, showing gene loci and CNV analyzes is redundant since the authors want to highlight the prognostic signature of various NSCLC clusters. However, I agree keeping the data in the manuscript and thanks for the explanation. ]

Response 2:  Many thanks for your understanding.